# Reduced mu opioid receptor availability in schizophrenia revealed with [$^{11}$C]-carfentanil positron emission tomographic Imaging

Abhishekh H. Ashok [1,2,3,4,8,9], Jim Myers[5,9], Tiago Reis Marques[1,2,3], Eugenii A. Rabiner [6,7] & Oliver D. Howes[1,2,3]*

Negative symptoms, such as amotivation and anhedonia, are a major cause of functional impairment in schizophrenia. There are currently no licensed treatments for negative symptoms, highlighting the need to understand the molecular mechanisms underlying them. Mu-opioid receptors (MOR) in the striatum play a key role in hedonic processing and reward function and are reduced post-mortem in schizophrenia. However, it is unknown if mu-opioid receptor availability is altered in-vivo or related to negative symptoms in schizophrenia. Using [$^{11}$C]-carfentanil positron emission tomography (PET) scans in 19 schizophrenia patients and 20 age-matched healthy controls, here we show a significantly lower MOR availability in patients with schizophrenia in the striatum (Cohen's $d = 0.7$), and the hedonic network. In addition, we report a marked global increase in inter-regional covariance of MOR availability in schizophrenia, largely due to increased cortical-subcortical covariance.

[1] Psychiatric Imaging Group, MRC London Institute of Medical Sciences, Hammersmith Hospital, Imperial College London, London, UK. [2] Faculty of Medicine, Psychiatric Imaging Group, Institute of Clinical Sciences (ICS), Imperial College London, London, UK. [3] Institute of Psychiatry, Psychology and Neuroscience, Kings College London, London, UK. [4] Cambridge University Hospitals NHS Foundation Trust, Cambridge, UK. [5] Faculty of Medicine, Imperial College London, London, UK. [6] Invicro, London, UK. [7] Department of Neuroimaging, Institute of Psychiatry, Psychology and Neuroscience, Kings College London, London, UK. [8] Present address: Department of Radiology, University of Cambridge, Cambridge, UK. [9] These authors contributed equally: Abhishekh H. Ashok, Jim Myers. *email: oliver.howes@lms.mrc.ac.uk

Schizophrenia affects about one in 100 people and is a leading cause of disease burden[1]. Patients with schizophrenia commonly experience negative symptoms, which include amotivation, anhedonia and asociality. Moreover, negative symptoms are associated with poor functional outcomes[2]. Antipsychotics only marginally improve negative symptoms[3], and there are no currently licensed treatments specifically for negative symptoms. In view of this, and the marked impact of negative symptoms on recovery, there is considerable interest in identifying potential treatment targets for these symptoms. However, the neurobiological mechanisms underlying negative symptoms remain unclear[4].

Impairments in hedonic function are thought to be a key mechanism underlying negative symptoms[2], conceptualised as reduced motivation to pursue reward (anticipatory anhedonia) and reduced hedonic response to reward itself (consummatory anhedonia). Studies assessing these features in schizophrenia have reported deficits in either consummatory[5], or anticipatory[6], or both components of hedonic function[7].

Converging lines of evidence indicate that MOR play a central role in hedonic function. Specifically, MOR knock out mouse models show anhedonic phenotypes such as decreased conditioned social place preference[8], and reduced nose pokes for sucrose pellets[9]. Moreover, pharmacological MOR blockade induces conditioned place aversion[10] and reduces social novelty preference[11]. Deficits in both sucrose preference[12] and progressive ratio response paradigms[9] suggest that mu-opioid system is involved in both the anticipatory and consummatory components of the anhedonic response. In contrast, MOR stimulation increases motivation to seek reward[13,14], and increases food palatability[15]. Consistent with these findings, human studies report blockade of MOR decreases the pleasantness of palatable foods[16] and sexual stimuli[17], whilst pharmacological stimulation of the opioid system increases the hedonic evaluation and motivation for viewing rewarding images[18].

The MOR system also modulates social interaction. Specifically, MOR knockout pups emit fewer ultrasonic vocalisations when removed from their mother, suggesting reduced maternal attachment behaviour[19]. Human molecular imaging studies using [11C]-carfentanil, a MOR agonist tracer, have shown that MOR activation in the amygdala, insula, and striatum are linked to increased social acceptance and interaction[20]. In summary, both preclinical and human studies show a potential role of MOR in mediating anhedonia, amotivation and asociality.

Opioid dysregulation in schizophrenia was first suggested in the 1980s[21] based on findings of elevated beta-endorphin levels in the cerebrospinal fluid of schizophrenia patients[22–24] and clinical studies of naloxone, a MOR antagonist treatment showing a reduction of psychotic symptoms in some patients[25]. Postmortem studies have found elevated MOR mRNA levels in the frontal cortex of patients[26], but lower binding of the MOR selective agonist, [3H]-DAMGO, in the cingulate gyrus and caudate-putamen of schizophrenia patients who died by suicide compared to healthy controls and patients who had died by non-suicide causes[27]. The inconsistency in the findings of post mortem studies could be due to confounders such as the inclusion of more people who had died by suicide in the patient group given that brain MOR mRNA and protein levels have been found to be altered in suicide victims compared to healthy controls[27–29].

Despite the preclinical, human studies and evidence from postmortem and peripheral measures of the potential role of MOR in schizophrenia, there have not been, to our knowledge, any previous PET studies of MOR availability in vivo in schizophrenia. [11C]-carfentanil is a selective MOR tracer with over two orders of magnitude higher affinity for MOR than other receptors [Ki($\mu$) = 0.024 nM, Ki($\delta$) = 3.28 nM, Ki($\kappa$) = 43.1nM[30]], shows excellent reproducibility (variability < 10%, intraclass correlation coefficients >0.93 in test-retest studies)[31] and kinetic properties, making it a good tracer to evaluate the MOR in vivo in neuropsychiatric disorders[30].

Here we show that patients with schizophrenia have reduced MOR availability in striatum and brain regions implicated in hedonic responses compared to healthy controls. In addition, we report a highly significant global increase in MOR connection strength in schizophrenia patients relative to controls, largely due to increased cortical-subcortical connectivity.

## Results

**Subject charecteristics.** Twenty individuals with schizophrenia and 20 healthy control subjects were studied. One patient had a structural abnormality on the MRI scan and was excluded from further analysis. Demographic details for all participants are given in Table 1. All patients were taking an antipsychotic at the time of the scan (listed in Supplementary Table 1). There was no significant difference between groups in age, sex, radioactive dose and injected mass per body weight (μgm/kg) received. There were more smokers in the patient group compared to healthy control group, and patients smoked significantly more cigarettes per day than controls (number of cigarettes smoked per day (mean ± SEM): patients = 8.2 ± 2.1 vs. controls = 0.9 ± 0.6; $p < 0.001$).

There were significant group differences in anhedonia and BMI, with higher anhedonia ratings in patients (Table 1). There was no significant difference in the *rs1799971* genotype frequency between groups. The gene frequency in our sample is consistent with previous studies which reported a A:A genotype frequency of 75%[32].

**Region of interest analysis.** The Region of interest (ROI) analysis showed significantly lower MOR availability in the striatum of patients with schizophrenia relative to controls (patients vs. controls (mean ± SEM): 1.54 ± 0.06 vs. 1.7 ± 0.05, Cohen's $d = 0.7$ $t = -2.2$, df (37), $p = 0.037$) (Fig. 1). These changes were significant in the dorsal striatum (patients vs. controls (mean ± SEM): 1.35 ± 0.06 vs. 1.53 ± 0.05, $p = 0.03$), but no significant differences were seen in the ventral striatum (patients vs. controls (mean ± SEM): 2.6 ± 0.08 vs. 2.69 ± 0.07, $p = 0.45$). There was no correlation between striatal MOR availability and negative symptom severity (Supplementary Fig. 1; PANSS-negative symptom subscale- $r$: 0.07, $p = 0.78$, SANS-25 total score- $r$: −0.151, $p = 0.54$), or social, physical, anticipatory and consummatory anhedonia measures in patients and controls (all $p > 0.05$). Similarly, there was no association between dorsal or ventral striatal MOR availability and negative symptom or anhedonia severity (all $p > 0.05$). Secondary analysis revealed a significant effect of both group ($F$ (5, 222) = 334.5, $p < 0.05$) and ROI ($F$ (1, 222) = 5.654 $p < 0.05$) on $BP_{ND}$ measures in the hedonic network (Fig. 2). The group × ROI interaction was not significant ($F$ (5, 222) = 0.2167, $p > 0.05$).

In view of the significant group differences in BMI and smoking, we conducted an exploratory analysis of the relationship between them and MOR. This showed there was no significant correlation between striatal MOR availability and BMI in patients (Pearson $r = -0.35$, $p = 0.1$) or controls (Pearson $r = -0.33$, $p = 0.15$). There was no association between number of tobacco cigarette smoked per day and MOR availability in striatum (patients: $r = -0.047$ $p = 0.85$; controls: $r = 0.27$, $p = 0.25$).

The area under the curve of the standardised uptake values for the occipital cortex did not differ between the groups (Supplementary Fig. 2, $p > 0.05$), consistent with the assumption of no difference in reference tissue MOR signal between the groups. To explore if antipsychotic treatment could influence our MOR findings, we calculated the chlorpromazine equivalent (CPZ

**Table 1 Demographics of study subjects**

|  | Schizophrenia patients (n = 19) (mean ± SEM) | Controls (n = 20) | p-value |
|---|---|---|---|
| Age (years) | 35.1 ± 2.1 | 36.85 ± 2.7 | 0.61 |
| Gender (male/female) | 19/0 | 18/2 | 0.256 |
| Injected radioactivity (MBq) | 198.6 ± 9.1 | 198.5 ± 9.7 | 0.9 |
| Injected mass per body weight (µgm/kg) | 0.023 ± 0.003 | 0.024 ± 0.003 | 0.6 |
| BMI (kg/m$^2$) | 29.3 ± 1 | 25.2 ± 0.8 | 0.003* |
| Mean age at onset (years) | 23.05 ± 1.2 | n/a | n/a |
| Mean duration of illness (years) | 11.3 ± 2.2 | n/a | n/a |
| PANSS |  | n/a | n/a |
|  Positive | 14.5 ± 0.4 |  |  |
|  Negative | 21.4 ± 1 |  |  |
|  General | 26.5 ± 0.8 |  |  |
|  Total | 62.5 ± 1.9 |  |  |
| SANS-25 | 55.7 ± 5 | n/a | n/a |
| Revised social anhedonia scale | 17.6 ± 1.9 | 9.8 ± 1.7 | 0.004* |
| Revised physical anhedonia scale | 23.3 ± 2.8 | 13.3 ± 3 | 0.017* |
| Temporal experience pleasure scale |  |  |  |
|  Anticipatory pleasure scale | 38.9 ± 2.6 | 43.2 ± 2 | 0.2 |
|  Consummatory pleasure scale | 28.3 ± 2.3 | 35 ± 1.7 | 0.025* |
| Genotyping rs1799971- |  |  |  |
|  A:A | 16 | 15 | 0.9 |
|  G:A | 3 | 4 |  |
|  G:G | 0 | 1 |  |
| Calgary depression scale total score | 7.7 ± 1.5 | n/a | n/a |
| Number of cigarette smoked per day | 8.2 ± 2.1 | 0.9 ± 0.6 | 0.001* |

There were no significant differences in demographics other than for body mass index (BMI) and number of cigarettes smoked per day
n/a not applicable; *p < 0.05

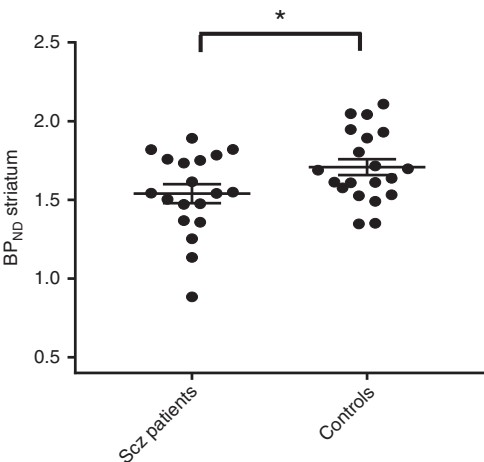

**Fig. 1** Striatal mu opioid receptor (MOR) availability in schizophrenia patients and controls showing means (horizontal line) and standard error (error bars). There was a significant reduction in the MOR availability in schizophrenia patients (cohen's d = 0.7; *p < 0.05)

equivalent) dose of antipsychotic treatment in all patients using the method described by Leucht et al.[33,34], and investigated if there was a relationship between antipsychotic dose and MOR. There was no association between striatal MOR availability and antipsychotic dose ($r = 0.06$, $p = 0.82$) (Supplementary Fig. 3). Furthermore, none of the antipsychotics has significant affinity for MOR (see Supplementary Table 2 for affinities).

There was no significant difference in the total grey matter volume between groups (patients vs. control: mean ± SEM (mm$^3$) 1171573 ± 16661 vs. 1217443 ± 21157, respectively, $p = 0.1$). Moreover, there were no significant differences in the volume of the striatum between groups (patients vs. controls: mean ± SEM (mm$^3$) 13019 ± 302 vs. 12937 ± 327, respectively, $p = 0.86$).

There was a significant difference between groups in the volumes of the regions in the hedonic network (Supplementary Table 3). However, there were no relationships between grey matter volume and MOR binding potential in the striatum or any of the other regions of interest (all $r$ −0.2 to 0.3, $p > 0.05$).

**Mu opioid receptor covariance network analysis**. Z-matrices for both patients and controls are shown in Fig. 3. Binding potentials throughout the cortex show moderate to high correlations with binding potentials in other cortical regions in both groups. However, qualitatively, the schizophrenia group shows more regions with high positive correlations relative to healthy controls (see Fig. 3). Quantitatively, we found a highly significant ($p < 0.00001$) increase in the connection strength of MOR $BP_{ND}$ in 1791 out of 7750 edges in schizophrenia patients relative to healthy controls with a threshold of $p < 0.05$ corrected for multiple comparisons using the network-based statistics algorithm (see Fig. 4a). This represents MOR correlations across the whole brain region, suggesting a global increased connectivity in patients compared to controls. To investigate the most significantly different networks between groups, we applied a more constrained primary threshold of $p < 0.001$ (corrected for multiple comparison using the network-based statistics algorithm). This identified increased MOR $BP_{ND}$ covariation in 99 out of 7750 edges in the schizophrenia patient group compared to controls in a cerebello-thalamo-cortical covariance network (see Fig. 4b).

## Discussion
MOR availability was reduced in vivo in the striatum with a large effect size (Cohen's $d = 0.7$) in schizophrenia patients compared to healthy controls, consistent with our main hypothesis. Our secondary analyses showed significantly reduced mu-opioid receptor availability across other brain regions involved in hedonic processes, comprising the orbitofrontal cortex, cingulate cortex, insular cortex, midbrain, and amygdala, in schizophrenia. Furthermore, the covariance network analysis showed

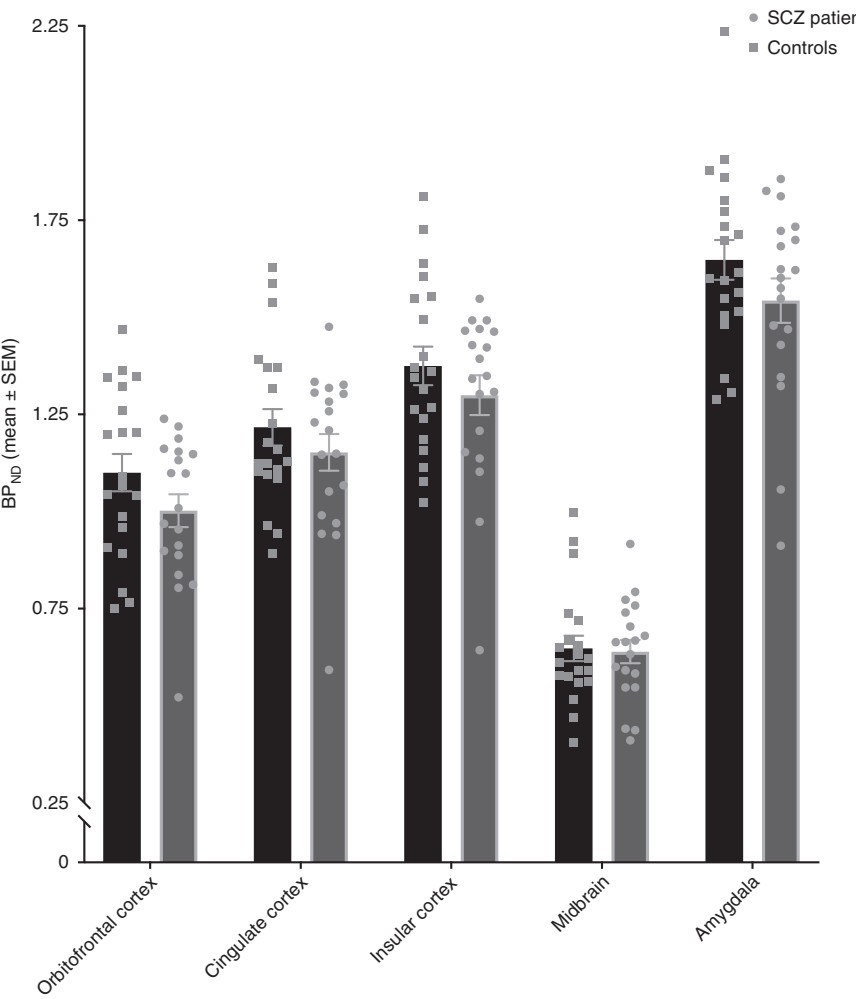

**Fig. 2** [$^{11}$C]-carfentanil binding potentials in the hedonic network of healthy controls and schizophrenia patients (mean ± SEM). There was a significant group effect on BP$_{ND}$ measures in the hedonic network, with lower levels in patients (*$p < 0.05$)

significantly elevated cortical-to-subcortical correlations between mu-opioid receptor levels in schizophrenia patients relative to controls. However, MOR availability was not linked to negative symptom or anhedonia severity ratings.

Our findings extend post-mortem findings of reduced MOR density measured with DAMGO, a MOR selective agonist, by showing reduced MOR availability in vivo and without the potential confound of suicide[27]. MOR mRNA levels have been reported to be elevated in the frontal cortex of patients with schizophrenia[26]. Studies in healthy volunteers have shown that MOR levels measured with PET are not closely correlated with mRNA levels, likely reflecting the fact the MOR undergoes post-transcriptional modification[35]. Thus, our findings taken with evidence of elevated MOR mRNA levels in the frontal cortex of patients[26], could suggest there is increased post-transcriptional modification and/or internalisation or breakdown of mu-opioid receptors in schizophrenia.

To our knowledge, this is the first study investigating MOR availability in vivo in schizophrenia. In our sample, all subjects were treated with antipsychotics. However, none of the antipsychotics taken by the patients has significant affinity for the MOR (all Ki values > 1000)[36–39]. Moreover, it is important to note that non-human primate studies show that neither halo-peridol nor olanzapine leads to appreciable alterations in the MOR availability, indicating that antipsychotic treatment does not significantly alter MOR[26]. Thus, it is unlikely that

antipsychotic treatment is a significant confounder. Nevertheless, several preclinical studies have shown that alteration in dopa-minergic activity can affect opioidergic neurotransmission[40]. Future studies in drug naïve subjects are needed to address this potential confounding effect.

We used the SRTM model with the occipital cortex as a reference region to estimate MOR availability. We used lower layers of the occipital cortex as the reference region as this has negligible MOR levels and has been validated by previous studies[41,42]. In addition, the area under the curve of the standardised uptake values for the occipital cortex, representing non-specific tracer accumulation in this region, did not differ between the groups, indicating that there are no significant group differences in tracer binding in this region.

Previous studies have reported altered MOR availability in morbidly obese subjects with BMI values of 38–42[43]. In our cohort, patients had a higher BMI (range 17.5–38) compared to controls (range 18.5–35), although not in the morbidly obese range. Moreover, there was no correlation between BMI and striatal MOR availability in our samples, suggesting the BMI difference is unlikely to account for our findings. Nevertheless, it would be useful in future studies to investigate this further.

Studies have shown that alcohol, opioid, stimulant, and tobacco smoking could affect the opioid signaling[44,45]. However, none of our patients and controls met criteria for present or past sub-stance use disorder, indicating this is unlikely to be a confound

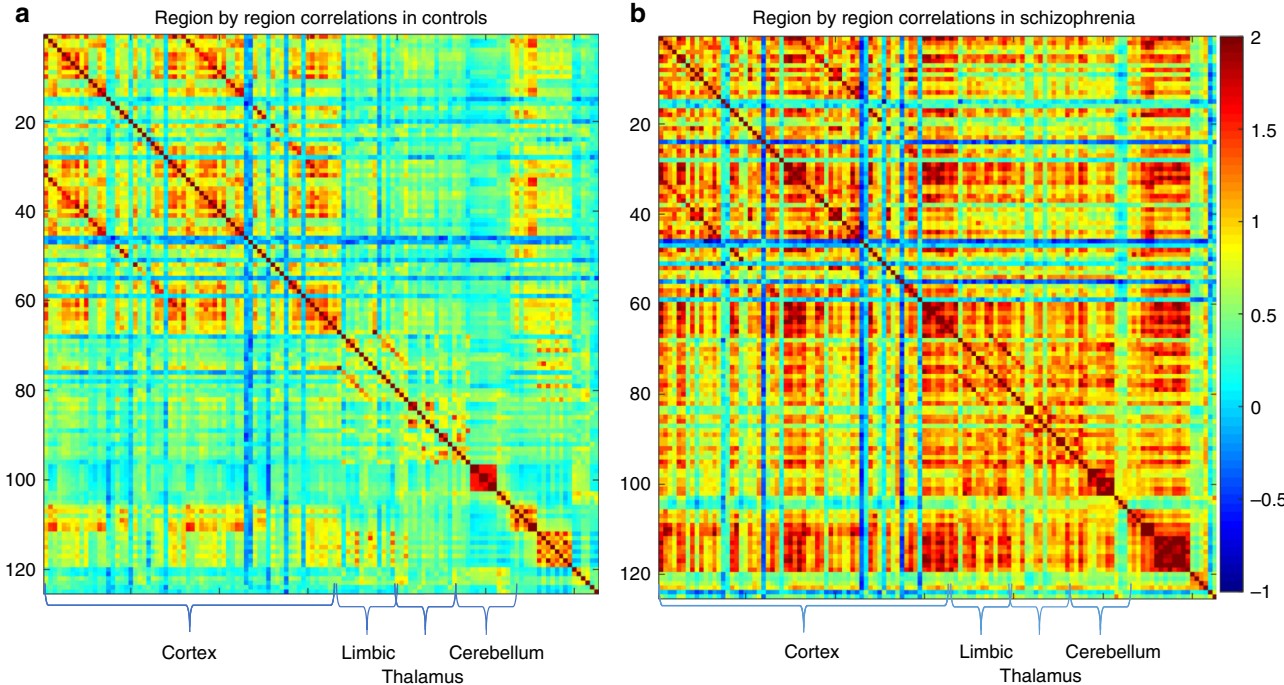

**Fig. 3** Matrices showing region-by-region correlations, represented as Z scores, throughout the whole brain (125 ROIs) by group. X and Y axes represent $BP_{ND}$ values across the 125 ROIs defined by the Clinical Imaging Centre (CIC) atlas. **a** shows data for controls. **b** shows data for patients. The colour bar represents the strength of correlation expressed as a Z-score (red = high positive correlation, blue = high negative correlation)

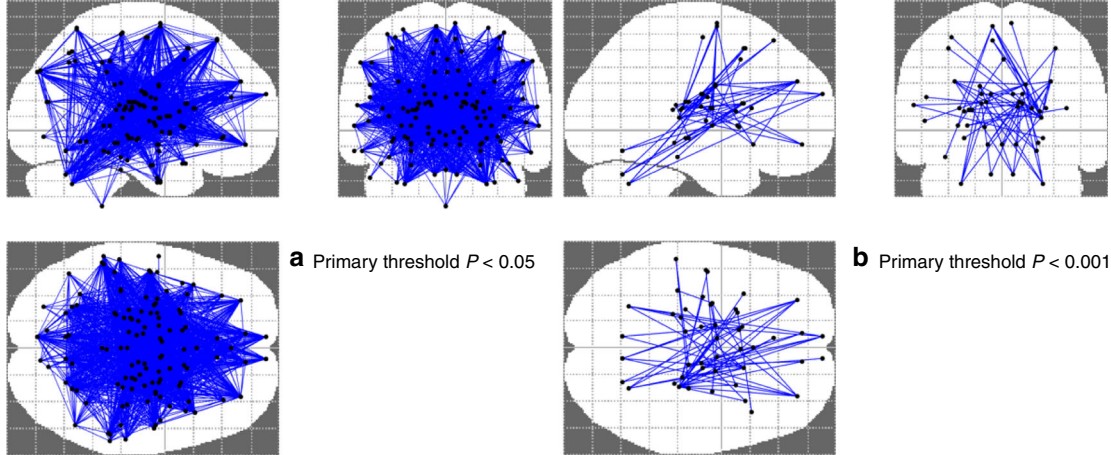

**Fig. 4** Regions identified as showing significantly greater covariance in schizophrenia compared to healthy controls. Using a primary threshold of $p < 0.05$ (false discovery rate corrected), **a** global increase in connectivity was shown in schizophrenia (left-hand side). A more conservative ($p < 0.001$ false discovery rate corrected) threshold identified the most significant components to be in a cerebello-thalamo-cortical network (**b**, right-hand side)

for the MOR availability measurements. Although, patients smoked more cigarretes than controls, there was no correlation between MOR availability and a number of tobacco cigarettes smoked, suggesting smoking is not a major confound. However, given that tobacco smoking may affect opioid signalling[44], it is possible that group differences in cigarette smoking could have influenced our findings.

There was no significant difference in the total gray matter and striatal volume between patients and controls. Further, there was no association between gray matter volume and $BP_{ND}$, suggesting partial volume effects are unlikely to be a significant confounder. However, there was a difference in the volumes between groups in the regions in the hedonic network, which could indicate that partial volume effects contribute to our findings in these regions. Notwithstanding this, we did not find any significant

relationships between gray matter volume in the striatum or any other region of interest, indicating that partial volume effects are unlikely to have had a major effect on our findings.

Previous post-mortem studies have reported no alteration in delta and kappa opioid receptors in schizophrenia[26]. Combined with the high selectivity of [11C]-carfentanil for the MOR (Ki = 0.0024 nM) over delta opioid (Ki = 3.28 nM) and kappa opioid receptors (Ki = 43.1 nM), our data indicate a lower MOR availability in schizophrenia patients[46].

The reduced $BP_{ND}$ could be due to receptor down-regulation, or neuronal loss[30], whilst our finding of increased cross-correlations between MOR levels across the brain in schizophrenia suggests a global dysregulation. The downregulation of the MOR could be due to long-term overstimulation of MOR by beta-endorphins and Leu-enkephalin, which both have high

affinity for MOR (Ki values of 0.3 and 1, respectively)[47]. This is consistent with the findings of elevated beta-endorphin[23] and enkepalin[48] levels in CSF from schizophrenia patients, although post mortem studies have not reported alteration in enkephalin mRNA levels in schizophrenia[26,49]. MOR activation inhibits release of GABA in the cortex[50] and, in the striatum, leads to dopamine release[51] and is associated with increased dopamine synthesis capacity[52]. Thus, increased beta-endorphin levels could lead to reduced cortical inhibition, leading to cognitive impairments, and striatal dopamine disinhibition in schizophrenia, leading to psychotic symptoms due to dopamine dysregulation[53]. However, as we did not measure beta-endorphin levels or dopamine function, it needs to be determined if lower MOR availability in the striatum in schizophrenia is due to elevated beta-endorphin levels or linked to dopamine dysfunction. [$^{11}$C]-carfentanil is sensitive to endogenous opioid levels, which compete with the tracer to reduce its binding to the MOR[42,54]. Thus, the reduction in MOR $BP_{ND}$ could be due to either reduced receptor availability or increased endogenous opioid release, or a combination of both. Future studies, using pharmacological challenges that release endogenous opioids, such as amphetamine, would be useful to determine if the lower levels of [$^{11}$C]-carfentanil binding we observed are due to altered endogenous opioid release or reduced MOR levels[41,42].

It is surprising that there was no significant association between MOR availability and either total negative symptoms or anhedonia severity (social, physical, anticipatory and consummatory anhedonia), given the lower MOR availability in patients. Our secondary analyses found that MOR availability was significantly lower in the dorsal but not ventral striatum in patients relative to controls, suggesting that the finding of lower striatal MOR in patients was driven by differences in the dorsal striatum. Striatal MOR blockade reduces the motivation to seek food[13] and sexual pleasure in animals[55]. In addition, there is some evidence that this particularly involves the dorsal striatum, including findings that endogenous opioids released in the dorsal striatum during food consumption are associated with motivation to eat but not with the hedonic orofacial response to food[13] and blockade of MOR in the dorsal striatum abolished formation of partner preference without evoking partner aversion[55]. In contrast, ventral striatal MOR blockade has generally been linked to anhedonia[56]. Thus, these findings indicate that our results of lower dorsal striatal MOR availability may contribute to the amotivation rather than anhedonic component of negative symptoms in schizophrenia, and the lack of major differences in ventral striatal MOR availability could indicate that another mechanism underlies anhedonia seen in schizophrenia. However, as we did not measure motivation, the association between dorsal striatal MOR and amotivation requires testing in patients. Alternatively, the lack of association between negative symptoms and MOR could indicate they are linked to other neurotransmitter dysfunction rather than MOR[2,57,58], or, given that our patients had been ill for a number of years, that MOR alterations underlie the development of negative symptoms but not their maintenance in chronic patients.

The mean PANSS in our cohort was 60 and the mean PANSS-negative symptom score in our cohort was 21, and the highest was 30 (total possible score = 49). The total severity rating is lower than typically reported in studies of acute relapses, but is consistent with recent randomised control trials of treatments for negative symptoms, where mean PANSS total scores were 47–80 and mean negative scores were 17–22[59–61]. Consistent with these studies and recommendations for studies of negative symptoms, we recruited subjects with predominant negative symptom without acute positive symptoms (no more than PANSS positive subscale score of 4) as these can confound the assessment of negative symptoms[62].

Thus, our study, in common with others in the literature[59–61], largely recruited patients with moderate symptom severity, which could affect generalisability to patients with more severe symptoms.

It has been suggested that negative symptoms can be considered as primary, that is intrinsic to the disease process, or secondary to other factors such as acute psychosis, comorbid depression, antipsychotic effects or other factors[63]. We excluded marked psychosis and comorbid depression in our sample, indicating that negative symptoms are unlikely to be due to these factors. We also applied the deficit schizophrenia criteria, which are used to identify patients with schizophrenia showing primary and enduring negative symptoms[64]. This suggests the negative symptoms in our patient sample are likely to be predominantly primary in nature. However, we can not exclude some contribution from secondary factors in the negative symptoms in our patients, which, if these have alternative mechanisms, may have reduced our ability to detect a relationship between MOR and negative symptoms. Future longitudinal studies in early course, untreated patients would be helpful to confirm our findings in patients at the onset of negative symptoms.

The application of the network-based statistical analysis to MOR is, to our knowledge, novel in both healthy volunteers and schizophrenia. Nevertheless, we applied an approach that is well-established in the fMRI literature and has been applied to PET studies of other receptors[65–67], a conservative p-value threshold ($p < 0.001$) and false discovery rate correction to control the type-1 error rate. This analysis found stronger [$^{11}$C]-carfentanil $BP_{ND}$ correlations across cerebello-thalamo-cortical regions in patients compared to healthy control. There are two main mechanisms that could account for this. One is it could be a consequence of increased CSF beta-endorphin levels in schizophrenia[23]. Beta-endorphin synthesised in pro-opiomelanocortin (POMC) neurons in the hypothalamus diffuses through cerebrospinal fluid to act on MOR throughout the brain by volume transmission[68]. Thus, given that [$^{11}$C]-carfentanil is displaced by endogenous MOR ligands, increased brain beta-endorphin levels in schizophrenia could lead to reduced variability of MOR, as is seen with exogenous opiate block, which will increase inter-regional correlations as the range of possible $BP_{ND}$ values is lower[69]. Alternatively, the higher inter-regional correlations could be due to altered genetic regulation of MOR expression. Supporting this, a preclinical study has shown that deletion of the mu opioid receptor gene (Oprm1) results in disrupted whole brain resting state functional connectivity[70]. Notwithstanding this evidence, it is important to recognise that the physiological significance of the increased MOR covariance network in schizophrenia remains to be determined. Combined MOR and functional imaging studies in patients and healthy volunteers would be useful to test this.

In conclusion, mu opioid receptor availability is reduced in the striatum and other brain regions involved in hedonic processes and shows increased cortical-subcortical correlations in schizophrenia. However, it is not associated with negative symptom severity or anhedonia measures. These findings provide in vivo evidence for altered brain opioid signaling in schizophrenia.

## Methods

**Subject recruitment**. The study protocol was approved by London-Camberwell St Giles Research Ethics Committee and approval to administer radioactive material was granted by Administration of Radioactive Substances Advisory Committee (ARSAC, UK). All participants provided written informed consent to participate after receiving a description of the study.

We recruited 20 patients with schizophrenia from secondary mental health services. All patients met DSM-IV criteria for schizophrenia. They were required to have a minimum score ≥4 on atleast one domain of positive and negative symptom scale (PANSS) negative symptom sub-scale[71] OR two or more negative symptoms with a score ≥3 on the PANSS-negative symptom sub-scale to ensure current negative symptoms. All patients were required to meet criteria for deficit syndrome defined, in accordance with guidelines[64,72], as (a) the presence of at least two out of

six of the following negative symptoms: restricted affect (referring to observed behaviours rather than to the patient's subjective experience); diminished emotional range (i.e., reduced range of the patient's subjective emotional experience); poverty of speech; curbing of interests; diminished sense of purpose; diminished social drive; and (b) a combination of two or more of the above symptoms have been present for the preceding 12 months and were always present during periods of clinical stability; and (c) the above symptoms are not secondary to other factors, including anxiety, drug effects, psychotic symptoms, mental retardation, or depression; and (d) the patient meets DSM criteria for schizophrenia. The secondary factors associated with negative symptoms were excluded based on clinical interview. All patients were required to be on a stable dose of an antipsychotic for at least four weeks before the scan (see Supplementary Table 1 for a list of treatments).

Twenty healthy volunteers were recruited from the same local catchment area through public advertisement. Inclusion criteria included no psychiatric morbidity as assessed by the Structured Clinical Interview for DSM IV (SCID), and no family history of psychosis. Exclusion criteria for all subjects were: history or current substance use disorder (other than to tobacco) as assessed by clinical interview, history of head injury or neurological abnormality, present or recent (1 month) use of opiates, antidepressants or other psychoactive medications including antiepileptics, or significant physical comorbidity (minor self-limiting illnesses were permitted) as assessed by history and physical examination and contraindications to PET or MRI scanning.

Subjects underwent a screening assessment, which included medical and psychiatric history as well as the history of alcohol, tobacco and other substance use and a physical examination. A urine drug screen was carried out on the scan day to exclude psychoactive drug use. Negative symptom severity was assessed using the scale for the Assessment of Negative Symptoms (SANS)[73]. In addition, the physical and social anhedonia rating scale[74], and temporal experience of pleasure scale (TEPS) were administered to the participants to assess anhedonia. The Calgary depression scale was used to assess depression[75].

**Genotyping**. Previous studies have shown that carriers of the OPRM1 G allele (rs1799971) show reduced [11C]-carfentanil binding[44,76]. In view of this, venous blood samples were taken for genotyping for the OPRM1 A118G polymorphism and were analysed by LGC Limited (Middlesex, UK). DNA was extracted, normalised and underwent SNP-specific KASP™ assay mix. Loci with a call rate < 90% were not included. Subjects were categorised as a G-allele carrier (G: A/ G: G) or not (A: A).

**Structural MRI acquisition**. High resolution T1 weighted volumes were acquired using a 3T MR scanner (Magneton Trio Syngo MR B13 Siemens 3T; Siemens AG, Germany) and a magnetisation prepared rapid gradient echo (MPRAGE) sequence (TR = 2300 ms, TE = 2.98 ms, TI = 900 ms, flip angle = 9°, field of view = 256 mm, image matrix = 240 × 256) with a resolution of 1 mm isotropic. For the volume, 160 abutting straight sagittal slices were collected in an interleaved right to left manner, resulting in whole head coverage. Parallel imaging using Generalized Auto calibrating Partially Parallel Acquisition (GRAPPA) with an acceleration factor of 2 was performed.

**PET acquisition**. [11C]-carfentanil, a selective MOR agonist, was synthesised by labelling its des-methyl precursor (4-Piperidinecarboxylic acid, 4-[(1-oxopropyl) phenylamino]-1-(2-phenylethyl), sodium salt; ABX Advanced Biochemical Compounds, Radeberg, Germany/ Pharmasynth, Tartu, Estonia), with carbon-11 using a modification of a previously described method[77] on a semi-automated Modular Lab Multifunctional Synthetic Module (Eckert & Ziegler, Berlin, Germany). The final product was reformulated in sterile 0.9% saline containing ∼10% ethanol (v/v) and satisfied quality control criteria for specific activity and purity before being injected intravenously as a slow bolus over ∼20 s. Following a transmission CT scan, a maximum of 300MBq of [11C]-carfentanil was administered. PET emission data were collected for 90 min in 26 frames (8 × 15 s, 3 × 60 s, 5 × 120 s, 5 × 300 s and 5 × 600 s, to a total of 5400 s). PET scans were acquired on a Siemens HiRez 6 PET/computed tomography scanner (Siemens Healthcare, Erlangen, Germany).

**PET image analysis**. Image pre-processing and PET modeling were carried out using MIAKAT™ software (www.miakat.org). Dynamic PET data were corrected for attenuation and scatter, and for motion by frame-by-frame realignment to frame 16. Each scan was rigid-body coregistered to the structural MRI. ROIs were defined using a neuroanatomical atlas[78], applied to the PET image by non-linear deformation parameters derived using unified segmentation of the structural MRI using statistical parametric mapping software (SPM 12). The template and atlas fits were confirmed visually for each participant. [11C]-carfentanil binding potential (BP$_{ND}$) values were quantified using the simplified reference tissue model (SRTM) with occipital lobe grey matter as the reference[41,79]. This approach shows good agreement on comparison with the arterial input function derived volume of distrbution[31,46], and the occipital cortex has negligible MOR availability[42,80–83].

We focussed on striatum as our primary region of interest because it plays a key role in the hedonic response, is rich in MOR[84], and studies in schizophrenia show striatal hypoactivation in response to reward[85], and reduced striatal MOR ligand

binding post-mortem[27]. In addition, studies have shown that a hedonic network, comprising the orbitofrontal cortex, cingulate cortex, insular cortex, midbrain, and amygdala as well as the striatum, is activated in response to hedonic stimuli[86], including to social reward[20,87]. Moreover, a meta-analysis of fMRI studies which focussed on neural correlates of anhedonia identified hypo-activation of the cingulate cortex, orbitofrontal cortex, as well as the striatum, during reward tasks in schizophrenia relative to controls[88]. In view of this, we conducted a secondary analysis to test the hypothesis that MOR availability was reduced in schizophrenia in the hedonic network consisting of the orbitofrontal cortex, cingulate cortex, insular cortex, midbrain, and amygdala.

To assess if [11C]-carfentanil uptake in the reference tissue differed between groups, and thus affected global BP$_{ND}$ calculations, standardised uptake values (SUV) were calculated by dividing the tissue radioligand uptake (calculated using the 10–90 min summed image) by injected dose per body weight.

The gray matter volume of the whole brain and the volumes of the ROIs were obtained from the individual's structural MRI scans after tissue segmentation as follows. The CIC atlas was non-linearly warped into subject space, and normalised to the subject's T1 weighted MRI images using SPM12 (SPM; https://www.fil.ion. ucl.ac.uk/spm/software/spm12/). After tissue segmentation also using SPM12, the grey matter volume for each ROI in the CIC atlas was then extracted as the volume of each ROI weighted by the grey matter probability.

**MOR covariance analysis**. In addition to regional alterations, it remains unknown if global brain MOR organisation is altered in schizophrenia. To assess this, we performed a correlation analysis for the whole brain and compared this for the two groups[89]. Within each group, the correlation coefficients between [11C]-carfentanil BP$_{ND}$ at each ROI with all other ROIs (125 ROIs defined in the Clinical Imaging Centre atlas[78]) were calculated and z values derived using the Fisher z-transformation to derive a global correlation matrix. This matrix was considered as a covariance network, where nodes are the ROIs and interregional correlations are the edges. The matrix consisted of 7750 edges. To determine the difference in the strength of connectivity of the edges between patient and control groups, each edge within these networks was compared between groups using permutation testing in MATLAB (100,000 permutations of group labels). To correct for the large number of partially dependent comparisons, the network-based statistic was used, with a primary threshold of alpha = 0.05[90]. To identify primary contributors to the effect between groups, we conducted a further analysis using a more conservative threshold, alpha = 0.001, consistent with the approach used in previous imaging studies using network-based statistics[35,89].

**Statistical analysis**. Statistical analysis was performed with SPSS (version 20) for MAC OS X and Graph pad prism version 7.04. Normality of distribution was tested using the Shapiro-Wilk test. The main hypothesis that there was a group difference in the MOR availability in the striatum was tested using an independent sample t-test. To determine whether there was an effect of group on BP$_{ND}$ values of the hedonic network, we performed two way-ANOVA with BP$_{ND}$ as the dependent variable and group (patient or control) as the independent variable. We examined correlations between PET and clinical data using Pearson's r. All data are presented as mean ± SEM, and the level α was set for all comparisons at P < 0.05.

**Reporting summary**. Further information on research design is available in the Nature Research Reporting Summary linked to this article.

## Data availability

All data and code used in the production of this manuscript are available on request.

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

## Acknowledgements

This study was funded by grants MC-A656-5QD30 from the Medical Research Council-UK, 666 from the Maudsley Charity, 094849/Z/10/Z from the Brain and Behavior Research Foundation, and Wellcome Trust to Dr Howes and King's College London scholarship to Dr Ashok. Dr Myers is supported by the Rosetrees Trust and Stoneygate Trust. Dr Rabiner wishes to acknowledge the support and funding by the National Institute for Health Research (NIHR) Biomedical Research Centre at South London and Maudsley NHS Foundation Trust and King's College London.

## Author contributions

A.H.A. had full access to all the data in the study and takes responsibility for the integrity of the data and the accuracy of the data analysis. Study concept and design: A.H.A, O.D.H. Acquisition, analysis, or interpretation of data: all authors. Drafting of the paper: all authors. Critical revision of the paper for important intellectual content: all authors. Statistical analysis: A.H.A., J.M. Administrative, technical, or material support: A.H.A. Study supervision: E.A.R., O.D.H.

## Competing interests

A.H.A. conducts research funded by the Medical Research Council (UK) and King's College London. O.D.H. conducts research funded by the Medical Research Council (UK), the National Institute of Health Research (UK) and the Maudsley Charity. O.D.H. has received investigator-initiated research funding from and/or participated in advisory/speaker meetings organised by Astra-Zeneca, BMS, Eli Lilly, Jansenn, Lundbeck, Lyden-Delta, Servier and Roche. Neither O.D.H. nor his family have been employed by or have holdings/a financial stake in any biomedical company. E.A.R. is an employee of Invicro, a Konica-Minolta company, and has a financial stake in GlaxoSmithKline. The funders had no role in the design and conduct of the study; collection, management, analysis, and interpretation of the data; preparation, review, or approval of the paper; and decision to submit the paper for publication. The remaining authors have no competing interests.
