## [Peer Review File · Nature Communications]

Reviewers' comments:

Reviewer #1 (Remarks to the Author):

This manuscript examines MOR BPND as measured with [11C]carfentanil and PET in a sample of 20 patients diagnosed with schizophrenia and age and sex-matched controls. The hypothesis behind the study is that alterations in MOR-mediated neurotransmission may underlie negative symptoms or anhedonia present in schizophrenia.

The authors find generalized reductions in MOR binding that were not associated with clinical characteristics (e.g., negative symptoms, anhedonia).

Major comments follow:

(1) It is suggested that reductions in MOR BPND are associated with reductions in endogenous opioid neurotransmission. However, this measure is a composite of receptor binding and neurotransmitter release. Therefore the authors do not present conclusive data regarding the activity of this neurotransmitter system simply using baseline measures.

(2) The measure of MOR connectivity has no physiological significance. Endogenous opioid projections are organized in short regulatory networks (enkephalins) or relatively longer projections (beta-endorphin). MOR's themselves do not represent "networks".

(3) The effects are fairly generalized and could represent partial volume averaging because of reductions in gray matter volume. In the absence of examination of gray matter volumes in the areas where the deficits in MOR BPND, it is unknown whether those deficits are due to the effects of atrophy.

(4) The effects of medications are not contemplated. There are known interactions between D2 and opioid systems which may induce the effects described and therefore may not be related to the diagnosis of schizophrenia.

Reviewer #2 (Remarks to the Author):

Review of the opioid receptor system in schizophrenia

This is an interesting, original and well-written paper examining for the first time - using a PET ligand - the availability of mu opioid receptors (MOR) in schizophrenia patients vs. healthy subjects. The authors convincingly suggest that the opioid system may be involved in schizophrenia, as they suggest, specifically regarding hedonic aspects of the illness and possibly negative symptoms. The PET scan is done with an appropriate ligand where patients with schizophrenia (N= 19) are compared to healthy subjects (n=20) without the presence of psychosis family history in the controls. The specific focus is on the striatum and the hedonic network in the brain. The main result is that in the striatum a lower binding potential in the patients is found as compared to the controls although no correlation is found with negative or (an)hedonic symptoms. The finding is interesting and novel but several aspects need to be improved in the paper. Specifically the introduction - although appropriate for the explanation and the background of the involvement of the opioid receptor in schizophrenia - does not mention the PET imaging ligand at all; no background is given, no argument why this ligand is appropriate, neither any data on previous studies using this ligand in schizophrenia, healthy subjects or any other psychiatric illness. This is a severe oversight of the authors. The selection of the patient is also rather peculiar. The severity of negative symptoms is actually pretty minimal though the study focuses on that. There is also no effort to make sure these are primary and not secondary negative symptoms. Indeed these patients are hardly ill at all with a total PANS score of about 60. Another issue is the areas that are being studied with PET. The striatum is mentioned but it most likely is

that the ventral striatum should have been targeted. Is it impossible to distinguish between the ventral and dorsal striatum using PET? If so, that should be made explicit. It is important that the antipsychotics that have been used in the patients is made explicit in the body of the paper even though the antipsychotics mentioned may not have an effect on the MOR. The discussion is well written and especially the limitations are appropriately discussed. There is an issue that tobacco smoking may affect opioid signaling. Although both groups were checked for drugs like alcohol and opioid and stimulants, it is very unlikely that smoking behavior is going to be similar in both groups. The smoking behavior of patients and controls should be made explicit including the number of cigarettes smoked in the patients - which most likely will be considerably higher than in the controls is the expectation. However, all in all, this is an interesting initial study using PET imaging of the mu opioid receptor in schizophrenia. It certainly opens new avenues for research and possibly even treatment.

Reviewer #1 (Remarks to the Author):

This manuscript examines MOR BP_{ND} as measured with [¹¹C]carfentanil and PET in a sample of 20 patients diagnosed with schizophrenia and age and sex-matched controls. The hypothesis behind the study is that alterations in MOR-mediated neurotransmission may underlie negative symptoms or anhedonia present in schizophrenia.

The authors find generalized reductions in MOR binding that were not associated with clinical characteristics (e.g., negative symptoms, anhedonia).

Major comments follow:

(1) It is suggested that reductions in MOR BP_{ND} are associated with reductions in endogenous opioid neurotransmission. However, this measure is a composite of receptor binding and neurotransmitter release. Therefore the authors do not present conclusive data regarding the activity of this neurotransmitter system simply using baseline measures.

>> We apologise for implying this and have amended the discussion to make clear this is an interpretation that needs further testing as follows (page 16):

[¹¹C]Carfentanil is sensitive to endogenous opioid levels, which compete with the tracer to reduce its binding to the MOR. (Mick et al., 2014; Quelch et al., 2014) Thus, the reduction in MOR BP_{ND} could be due to either reduced receptor availability or increased endogenous opioid release, or a combination of both. Future studies, using pharmacological challenges that release endogenous opioids, such as amphetamine or acetate, would be useful to determine if the lower levels of [¹¹C]Carfentanil binding we observed are due to altered endogenous opioid release or reduced MOR levels (Colasanti et al., 2012; Mick et al., 2014; Quelch et al., 2014)(Ashok et al in sub).

(2) The measure of MOR connectivity has no physiological significance. Endogenous opioid projections are organized in short regulatory networks (enkephalins) or relatively longer projections (beta-endorphin). MOR's themselves do not represent "networks".

>> We apologise for the lack of clarity here. We did not mean to imply that MOR are networks in their own right, and apologise if our terminology here was confusing. As the reviewer notes, opioidergic projections are organised in longer projections, and, in addition, beta-endorphins are shown to have behavioural effects by volume transmission through CSF (Veening and Barendregt, 2015; Veening et al., 2012). We have amended the paper to make clear that this is a covariance network and to discuss these issues and make clear that the physiological significance requires determination in the Discussion (page: 18) as follows:

The application of the network-based statistical analysis to MOR is, to our knowledge, novel in both healthy volunteers and schizophrenia. Nevertheless, we applied an approach that is well-established in the fMRI literature and has been applied to PET studies of other receptors (Cervenka et al., 2010; Erritzoe et al., 2010; Tuominen et al., 2014), a conservative p-value threshold (p<0.001) and false discovery rate correction to control the type-1 error rate. This analysis found stronger [¹¹C]Carfentanil BP_{ND} correlations across cerebello-thalamo-cortical regions in patients compared to healthy control. There are two main mechanisms that could account for this. One is it could be a consequence of increased CSF beta-endorphin levels in schizophrenia (Brambilla et al., 1984). Beta-endorphin synthesised in pro-opiomelanocortin (POMC) neurons in the hypothalamus diffuses through cerebrospinal fluid to act on MOR throughout the brain by volume transmission (Veening and

Barendregt, 2015; Veening et al., 2012). Thus, given that [¹¹C]Carfentanil is displaced by endogenous MOR ligands, increased brain beta-endorphin levels in schizophrenia could lead to reduced variability of MOR, as is seen with exogenous opiate block, which will increase inter-regional correlations as the range of possible BP_{ND} values is lower (Weerts et al., 2008). Alternatively, the higher inter-regional correlations could be due to altered genetic regulation of MOR expression. Supporting this, a preclinical study has shown that deletion of the mu opioid receptor gene (Oprm1) results in disrupted whole brain resting state functional connectivity (Mechling et al., 2016). Notwithstanding this evidence, it is important to recognise that the physiological significance of the increased MOR covariance network in schizophrenia remains to be determined. Combined MOR and functional imaging studies in patients and healthy volunteers would be useful to test this.”

(3) The effects are fairly generalized and could represent partial volume averaging because of reductions in gray matter volume. In the absence of examination of gray matter volumes in the areas where the deficits in MOR BP_{ND}, it is unknown whether those deficits are due to the effects of atrophy.

>> We thank reviewer for raising this point. We have conducted further analyses of gray matter volumes as suggested to address this issue. There was no significant difference in the total gray matter volume between patients vs control: mean ± SD (mm³)= 1171573 ± 72626 vs 1217443 ± 94619 respectively, p= 0.1. Moreover, there was no significant difference in the volume of the striatum between patients and controls (mean ± SD striatal volume: patients vs controls: 13019 ± 1317 mm³ vs 12937 ± 1464 mm³ respectively, p = 0.86). There was significant difference between groups in the volumes of hedonic regions (Supplementary table 3). In addition, we investigated if there was a relationship between gray matter volumes and MOR BP_{ND} in the regions of interest. There were no relationships between grey matter volume and MOR binding potential in any of the regions of interest (all r -0.2 to 0.3, p>0.05). Thus we found no evidence to indicate that partial volume effects underlie our findings in our primary region of interest (the striatum) but they could influence findings in some of our secondary regions of interest.

We have amended the methods (page 24), results (page 7) and discussion (page 15) sections to present these new analyses and consider them as follows:

“Methods: The gray matter volume of the whole brain and the volumes of the ROIs were obtained from the individual’s structural MRI scans after tissue segmentation as follows. The CIC atlas was non-linearly warped into subject space, and normalised to the subject’s T1 weighted MRI images using SPM12 (SPM; <https://www.fil.ion.ucl.ac.uk/spm/software/spm12/>). After tissue segmentation also using SPM12, the gray matter volume for each ROI in the CIC atlas was then extracted as the volume of each ROI weighted by the gray matter probability.

Results: There was no significant difference in the total gray matter volume between groups (patients vs control: mean ± SD (mm³) 1171573 ± 72626 vs 1217443 ± 94619 respectively, p= 0.1). Moreover, there were no significant differences in the volume of the striatum between groups (patients vs controls: mean ± SD (mm³) 13019 ± 1317 mm³ vs 12937 ± 1464 mm³ respectively, p = 0.86). There was a significant difference between groups in the volumes of the regions in the hedonic network (Supplementary table 3). However, there were no relationships between grey matter volume and MOR binding potential in the striatum or any of the other regions of interest (all r -0.2 to 0.3, p>0.05).

Supplementary table 3: Volume differences between patients and controls in hedonic regions (in mm³).

Regions	Patients Mean ± SD	Controls Mean ± SD	t	df	Sig. (2-tailed)
Anterior cingulate cortex	31228 ± 2108	34003 ± 5115	-2.193	37	.035
Amygdala	3611 ± 306	3884 ± 384	-2.446	37	.019
Orbitofrontal cortex	26978 ± 3399	30174 ± 4384	-2.535	37	.016
Insula	12962 ± 1078	13959 ± 1671	-2.199	37	.034

Discussion: There was no significant difference in the total gray matter and striatal volume between patients and controls. Further, there was no association between gray matter volume and BP_{ND}, suggesting partial volume effects are unlikely to be a significant confounder. However, there was difference in the volumes between groups in the regions in the hedonic network, which could indicate that partial volume effects contribute to our findings in these regions. Notwithstanding this, we did not find any significant relationships between gray matter volume in the striatum or any other region of interest, indicating that partial volume effects are unlikely to have had a major effect on our findings.

(4) The effects of medications are not contemplated. There are known interactions between D2 and opioid systems which may induce the effects described and therefore may not be related to the diagnosis of schizophrenia.

>> We thank the reviewer for this comment. We apologise if the discussion of this issue was not clear. We have now extended the discussion of medication effects and added additional analyses on the association of MOR availability with Chlorpromazine equivalent in the results, as follows (page 7):

“To explore if antipsychotic treatment could influence our MOR findings, we calculated the chlorpromazine equivalent (CPZ equivalent) dose of antipsychotic treatment in all patients using the method described by Leucht et al (Leucht et al., 2016; Woods, 2003), and investigated if there was a relationship between antipsychotic dose and MOR. There was no association between striatal MOR availability and antipsychotic dose ($r=0.06$, $p=0.82$)(Supplementary figure 3). Furthermore, none of the antipsychotics has significant affinity for MOR (see supplementary table 2 for affinities).”

And in the discussion as follows (page 13):

“In our sample, all subjects were treated with antipsychotics. However, none of the antipsychotics taken by the patients has significant affinity for the MOR (all K_i values > 1000)(Abbas et al., 2009; Kalkman et al., 2001; Schotte et al., 1996; Shapiro et al., 2003). Moreover, it is important to note that non-human primate studies show that neither haloperidol nor olanzapine leads to appreciable alterations in the MOR availability, indicating that antipsychotic treatment does not significantly alter MOR (Volk et al., 2012). Thus, it is unlikely that antipsychotic treatment is a significant confounder.

Supplementary figure 3: There was no association between striatal MOR availability and antipsychotic dose, expressed as the chlorpromazine equivalent dose (CPZE, $r=0.06$, $p=0.82$).

Supplementary table 2: Affinity of antipsychotics to μ -Opioid receptor

Drug	K_i	Species	Brain region	Ligand used to determine MOR binding	Reference
Clozapine	1000	Rat	Forebrain	3H-Sufentanil	(Schotte et al., 1996)
Haloperidol	1000	Rat	forebrain	3H-Sufentanil	(Schotte et al., 1996)
Olanzapine	1000	Rat	forebrain	3H-Sufentanil	(Schotte et al., 1996)
Quetiapine	1000	Rat	forebrain	3H-Sufentanil	(Schotte et al., 1996)
Risperidone	1000	Rat	forebrain	3H-Sufentanil	(Schotte et al., 1996)
Sertindole	1000	Rat	forebrain	3H-Sufentanil	(Schotte et al., 1996)
Ziprasidone	1000	Rat	forebrain	3H-Sufentanil	(Schotte et al., 1996)
Zotepine	1000	Rat	forebrain	3H-Sufentanil	(Schotte et al., 1996)
Aripiprazole	>10,000	Human	cloned	3H-Diprenorphine	(Shapiro et al., 2003)
Amisulpride	>10,000	Human	cloned	3H-DAMGO	(Abbas et al., 2009)
lloperidone	>10,000	Human	cloned	3H-NALOXONE	(Kalkman et al., 2001)

CPZE= chlorpromazine equivalent daily dose.

Reviewer #2 (Remarks to the Author):

Review of the opioid receptor system in schizophrenia

This is an interesting, original and well-written paper examining for the first time - using a PET ligand - the availability of mu opioid receptors (MOR) in schizophrenia patients vs. healthy subjects. The authors convincingly suggest that the opioid system may be involved in schizophrenia, as they suggest, specifically regarding hedonic aspects of the illness and possibly negative symptoms. The PET scan is done with an appropriate ligand where patients with schizophrenia (N= 19) are compared to healthy subjects (n=20) without the presence of psychosis family history in the controls. The specific focus is on the striatum and the hedonic network in the brain. The main result is that in the striatum a lower binding potential in the patients is found as compared to the controls although no correlation is found with negative or (an)hedonic symptoms. The finding is interesting and novel but several aspects need to be improved in the paper.

Specifically the introduction – although appropriate for the explanation and the background of the involvement of the opioid receptor in schizophrenia - does not mention the PET imaging ligand at all; no background is given, no argument why this ligand is appropriate, neither any data on previous studies using this ligand in schizophrenia, healthy subjects or any other psychiatric illness. This is a severe oversight of the authors.

>> We thank the reviewer for raising this issue. We have now modified the introduction to include this information (page 4) as follows:

“Despite the preclinical, human studies and evidence from post-mortem and peripheral measures of the potential role of MOR in schizophrenia, there have not been, to our knowledge, any previous PET studies of MOR availability in vivo in schizophrenia. [¹¹C]-carfentanil is a selective MOR tracer with

over two orders of magnitude higher affinity for MOR than other receptors [$K_i(\mu)$ = 0.024nM, $K_i(\delta)$ =3.28nM, $K_i(\kappa)$ = 43.1nM (Henriksen and Willoch, 2008)], and shows excellent reproducibility (variability <10%, intraclass correlation coefficients >0.93 in test-retest studies)(Hirvonen et al., 2009) and kinetic properties, making it a good tracer to evaluate the MOR in vivo in neuropsychiatric disorders (Henriksen and Willoch, 2008).”

The selection of the patient is also rather peculiar. The severity of negative symptoms is actually pretty minimal though the study focuses on that. Indeed these patients are hardly ill at all with a total PANS score of about 60.

>> We apologise that this wasn't clearer and thank the reviewer for this comment. The total PANSS scores may partially reflect the requirement to meet the deficit criteria for negative symptoms (see below). We have now made this clear in the method (criteria for deficit syndrome- given below) and discussed the symptom severity in the paper as follows:

Discussion (page 17):

The mean PANSS in our cohort was 60 and the mean PANSS negative symptom score in our cohort was 21, and the highest was 30 (total possible score=49). The total severity rating is lower than typically reported in studies of acute relapses, but is consistent with recent randomized control trials of treatments for negative symptoms, where mean PANSS total scores were 47-80 and mean negative scores were 17-22 (Bugarski-Kirola et al., 2017; Chaudhry et al., 2012; Deakin et al., 2018). Consistent with these studies and recommendations for studies of negative symptoms, we recruited subjects with predominant negative symptom without acute positive symptoms (no more than PANSS positive subscale score of 4) as these can confound the assessment of negative symptoms (Chen et al., 2013; Kane et al., 1988; Tandon et al., 2000; Tandon et al., 1993). Thus, our study, in common with others in the literature (Bugarski-Kirola et al., 2017; Chaudhry et al., 2012; Deakin et al., 2018), largely recruited patients with moderate symptom severity, which could affect generalisability to patients with more severe symptoms.

There is also no effort to make sure these are primary and not secondary negative symptoms.

- We thank the reviewer for raising this point. We applied the Carpenter et al criteria for deficit syndrome to recruit patients whose symptoms are likely primary and not secondary to other factors. We apologise that this was not made clearer, and have amended the method and extended the discussion to consider these issues as follows:

Methods (page 20):

All patients were required to meet criteria for deficit syndrome defined, in accordance with guidelines (Carpenter et al., 1988; Kirkpatrick et al., 1989), as a) the presence of at least two out of six of the following negative symptoms: restricted affect (referring to observed behaviours rather than to the patient's subjective experience); diminished emotional range (i.e., reduced range of the patient's subjective emotional experience); poverty of speech; curbing of interests; diminished sense of purpose; diminished social drive; and b) a combination of two or more of the above symptoms have been present for the preceding 12 months and were always present during periods of clinical stability; and c) the above symptoms are not secondary to other factors, including anxiety, drug effects, psychotic symptoms, mental retardation, or depression; and d) the patient meets DSM criteria for schizophrenia. The secondary factors associated with negative symptoms were excluded based on clinical interview.

Discussion (page 17):

It has been suggested that negative symptoms can be considered as primary, that is intrinsic to the disease process, or secondary to other factors such as acute psychosis, comorbid depression, antipsychotic effects or other factors (Kirschner et al., 2017). We excluded marked psychosis and comorbid depression in our sample, indicating that negative symptoms are unlikely to be due to these factors. We also applied the deficit schizophrenia criteria, which are used to identify patients with schizophrenia showing primary and enduring negative symptoms (Carpenter et al., 1988). This suggests the negative symptoms in our patient sample are likely to be predominantly primary in nature. However, we can not exclude some contribution from secondary factors in the negative symptoms in our patients, which, if these have alternative mechanisms, may have reduced our ability to detect a relationship between MOR and negative symptoms. Future longitudinal studies in early course, untreated patients would be helpful to confirm our findings in patients at the onset of negative symptoms.

Another issue is the areas that are being studied with PET. The striatum is mentioned but it most likely is that the ventral striatum should have been targeted. Is it impossible to distinguish between the ventral and dorsal striatum using PET? If so, that should be made explicit.

We thank the reviewer for this excellent suggestion. There is evidence that both the dorsal and ventral striatum are involved in MOR signalling (DiFeliceantonio et al., 2012; Ward et al., 2006). In view of this, our primary analysis did not differentiate these sub-regions of the striatum. However, in light of the reviewer's suggestion we have now conducted additional analyses of both the ventral and dorsal striatum. We have now included following information in the results section (page 11) and discussed this as follows:

These changes were significant in the dorsal striatum (patients vs. controls (mean \pm SEM): 1.35 ± 0.06 vs. 1.53 ± 0.05 , $p = 0.03$), but no significant differences were seen in the ventral striatum (patients vs. controls (mean \pm SEM): 2.6 ± 0.08 vs. 2.69 ± 0.07 , $p = 0.45$). There was no correlation between striatal MOR availability and negative symptom severity (supplementary figure 1; PANSS-negative symptom subscale- $r = 0.07$, $p = 0.78$, SANS-25 total score- $r = -0.151$, $p = 0.54$), or social, physical, anticipatory and consummatory anhedonia measures in patients and controls (all $p > 0.05$). Similarly, there was no association between dorsal or ventral striatal MOR availability and negative symptom or anhedonia severity (all $p > 0.05$).

Discussion (page 16):

Our secondary analyses found that MOR availability was significantly lower in the dorsal but not ventral striatum in patients relative to controls, suggesting that the finding of lower striatal MOR in patients was driven by differences in the dorsal striatum. Striatal MOR blockade reduces the motivation to seek food (DiFeliceantonio et al., 2012) and sexual pleasure in animals (Burkett et al., 2011; Resendez et al., 2013). In addition, there is some evidence that this particularly involves the dorsal striatum, including findings that endogenous opioids released in the dorsal striatum during food consumption are associated with motivation to eat but not with the hedonic orofacial response to food (DiFeliceantonio et al., 2012) and blockade of MOR in the dorsal striatum abolished formation of partner preference without evoking partner aversion (Burkett et al., 2011; Resendez et al., 2013). In contrast, ventral striatal MOR blockade has generally been linked to anhedonia (Ward et al., 2006). Thus, these findings indicate that our results of lower dorsal striatal MOR availability may contribute to the amotivation rather than anhedonic component of negative symptoms in schizophrenia, and the lack of major differences in ventral striatal MOR availability could indicate that another mechanism

underlies anhedonia seen in schizophrenia. However, as we did not measure motivation, the association between dorsal striatal MOR and amotivation requires testing in patients.

It is important that the antipsychotics that have been used in the patients is made explicit in the body of the paper even though the antipsychotics mentioned may not have an effect on the MOR.

>> We have now included this information in the methods (page 20) and a complete list in the supplementary informations, as follows:

All patients were required to be on a stable dose of an antipsychotic for at least four weeks before the scan (Supplementary table 1).

The discussion is well written and especially the limitations are appropriately discussed.

There is an issue that tobacco smoking may affect opioid signaling. Although both groups were checked for drugs like alcohol and opioid and stimulants, it is very unlikely that smoking behavior is going to be similar in both groups. The smoking behavior of patients and controls should be made explicit including the number of cigarettes smoked in the patients - which most likely will be considerably higher than in the controls is the expectation.

Apologies for this oversight. We have now included this information in the demographic table and conducted additional analyses to test for a relationship between smoking (page 8) and MOR and discussed (page 14) this as follows.

“There were more smokers in the patient group compared to healthy control group, and patients smoked significantly more cigarettes per day than controls (mean (sd) number of cigarettes smoked per day: patients=8.2 ± 2.1 vs controls=0.9 ± 0.6; $p < 0.001$)

There was no association between number of tobacco cigarette smoked per day and MOR availability in striatum (patients: $r = -0.047, p = 0.85$; controls: $r = 0.27, p = 0.25$).”

Discussion

Although, patients smoked more cigarettes than controls, there was no correlation between MOR availability and a number of tobacco cigarettes smoked, suggesting smoking is not a major confound. However, given that tobacco smoking may affect opioid signalling (Ray et al., 2011; Scott et al., 2007), it is possible that group differences in cigarette smoking could have influenced our findings.

However, all in all, this is an interesting initial study using PET imaging of the mu opioid receptor in schizophrenia. It certainly opens new avenues for research and possibly even treatment.

Reference:

- Abbas, A.I., Hedlund, P.B., Huang, X.P., Tran, T.B., Meltzer, H.Y., Roth, B.L., 2009. Amisulpride is a potent 5-HT₇ antagonist: relevance for antidepressant actions in vivo. *Psychopharmacology* 205, 119-128.
- Brambilla, F., Facchinetti, F., Petraglia, F., Vanzulli, L., Genazzani, A.R., 1984. Secretion pattern of endogenous opioids in chronic schizophrenia. *The American journal of psychiatry* 141, 1183-1189.
- Bugarski-Kirola, D., Blaettler, T., Arango, C., Fleischhacker, W.W., Garibaldi, G., Wang, A., Dixon, M., Bressan, R.A., Nasrallah, H., Lawrie, S., Napieralski, J., Ochi-Lohmann, T., Reid, C., Marder, S.R., 2017. Bitopertin in Negative Symptoms of Schizophrenia-Results From the Phase III FlashLyte and DayLyte Studies. *Biological psychiatry* 82, 8-16.
- Burkett, J.P., Spiegel, L.L., Inoue, K., Murphy, A.Z., Young, L.J., 2011. Activation of mu-opioid receptors in the dorsal striatum is necessary for adult social attachment in monogamous prairie voles. *Neuropsychopharmacology : official publication of the American College of Neuropsychopharmacology* 36, 2200-2210.
- Carpenter, W.T., Jr., Heinrichs, D.W., Wagman, A.M., 1988. Deficit and nondeficit forms of schizophrenia: the concept. *The American journal of psychiatry* 145, 578-583.
- Cervenka, S., Varrone, A., Fransen, E., Halldin, C., Farde, L., 2010. PET studies of D₂-receptor binding in striatal and extrastriatal brain regions: Biochemical support in vivo for separate dopaminergic systems in humans. *Synapse (New York, N.Y.)* 64, 478-485.
- Chaudhry, I.B., Hallak, J., Husain, N., Minhas, F., Stirling, J., Richardson, P., Dursun, S., Dunn, G., Deakin, B., 2012. Minocycline benefits negative symptoms in early schizophrenia: a randomised double-blind placebo-controlled clinical trial in patients on standard treatment. *Journal of psychopharmacology (Oxford, England)* 26, 1185-1193.
- Chen, L., Johnston, J.A., Kinon, B.J., Stauffer, V., Succop, P., Marques, T.R., Ascher-Svanum, H., 2013. The longitudinal interplay between negative and positive symptom trajectories in patients under antipsychotic treatment: a post hoc analysis of data from a randomized, 1-year pragmatic trial. *BMC psychiatry* 13, 320.
- Colasanti, A., Searle, G.E., Long, C.J., Hill, S.P., Reiley, R.R., Quelch, D., Erritzoe, D., Tziortzi, A.C., Reed, L.J., Lingford-Hughes, A.R., Waldman, A.D., Schruers, K.R., Matthews, P.M., Gunn, R.N., Nutt, D.J., Rabiner, E.A., 2012. Endogenous opioid release in the human brain reward system induced by acute amphetamine administration. *Biological psychiatry* 72, 371-377.
- Deakin, B., Suckling, J., Barnes, T.R.E., Byrne, K., Chaudhry, I.B., Dazzan, P., Drake, R.J., Giordano, A., Husain, N., Jones, P.B., Joyce, E., Knox, E., Krynicki, C., Lawrie, S.M., Lewis, S., Lisiecka-Ford, D.M., Nikkheslat, N., Pariante, C.M., Smallman, R., Watson, A., Williams, S.C.R., Upthegrove, R., Dunn, G., 2018. The benefit of minocycline on negative symptoms of schizophrenia in patients with recent-onset psychosis (BeneMin): a randomised, double-blind, placebo-controlled trial. *The lancet. Psychiatry* 5, 885-894.
- DiFeliceantonio, A.G., Mabrouk, O.S., Kennedy, R.T., Berridge, K.C., 2012. Enkephalin surges in dorsal neostriatum as a signal to eat. *Current biology : CB* 22, 1918-1924.
- Erritzoe, D., Holst, K., Frokjaer, V.G., Licht, C.L., Kalbitzer, J., Nielsen, F.A., Svarer, C., Madsen, J., Knudsen, G., 2010. A nonlinear relationship between cerebral serotonin transporter and 5-HT_{2A} receptor binding: an in vivo molecular imaging study in humans. *The Journal of neuroscience : the official journal of the Society for Neuroscience* 30, 3391-3397.

- Galderisi, S., Merlotti, E., Mucci, A., 2015. Neurobiological background of negative symptoms. *European archives of psychiatry and clinical neuroscience* 265, 543-558.
- Galderisi, S., Mucci, A., Buchanan, R.W., Arango, C., 2018. Negative symptoms of schizophrenia: new developments and unanswered research questions. *The lancet. Psychiatry* 5, 664-677.
- Henriksen, G., Willoch, F., 2008. Imaging of opioid receptors in the central nervous system. *Brain : a journal of neurology* 131, 1171-1196.
- Hirvonen, J., Aalto, S., Hagelberg, N., Maksimow, A., Ingman, K., Oikonen, V., Virkkala, J., Nagren, K., Scheinin, H., 2009. Measurement of central mu-opioid receptor binding in vivo with PET and [¹¹C]carfentanil: a test-retest study in healthy subjects. *European journal of nuclear medicine and molecular imaging* 36, 275-286.
- Howes, O.D., McCutcheon, R., Owen, M.J., Murray, R.M., 2017. The Role of Genes, Stress, and Dopamine in the Development of Schizophrenia. *Biological psychiatry* 81, 9-20.
- Kalkman, H.O., Subramanian, N., Hoyer, D., 2001. Extended radioligand binding profile of iloperidone: a broad spectrum dopamine/serotonin/norepinephrine receptor antagonist for the management of psychotic disorders. *Neuropsychopharmacology : official publication of the American College of Neuropsychopharmacology* 25, 904-914.
- Kane, J., Honigfeld, G., Singer, J., Meltzer, H., 1988. Clozapine for the treatment-resistant schizophrenic. A double-blind comparison with chlorpromazine. *Archives of general psychiatry* 45, 789-796.
- Kirkpatrick, B., Buchanan, R.W., McKenney, P.D., Alphas, L.D., Carpenter, W.T., Jr., 1989. The Schedule for the Deficit syndrome: an instrument for research in schizophrenia. *Psychiatry research* 30, 119-123.
- Kirschner, M., Aleman, A., Kaiser, S., 2017. Secondary negative symptoms - A review of mechanisms, assessment and treatment. *Schizophrenia research* 186, 29-38.
- Leucht, S., Samara, M., Heres, S., Davis, J.M., 2016. Dose Equivalents for Antipsychotic Drugs: The DDD Method. *Schizophrenia bulletin* 42 Suppl 1, S90-94.
- Mechling, A.E., Arefin, T., Lee, H.L., Bienert, T., Reisert, M., Ben Hamida, S., Darcq, E., Ehrlich, A., Gaveriaux-Ruff, C., Parent, M.J., Rosa-Neto, P., Hennig, J., von Elverfeldt, D., Kieffer, B.L., Harsan, L.A., 2016. Deletion of the mu opioid receptor gene in mice reshapes the reward-aversion connectome. *Proceedings of the National Academy of Sciences of the United States of America* 113, 11603-11608.
- Mick, I., Myers, J., Stokes, P.R., Erritzoe, D., Colasanti, A., Bowden-Jones, H., Clark, L., Gunn, R.N., Rabiner, E.A., Searle, G.E., Waldman, A.D., Parkin, M.C., Brailsford, A.D., Nutt, D.J., Lingford-Hughes, A.R., 2014. Amphetamine induced endogenous opioid release in the human brain detected with [(1)(1)C]carfentanil PET: replication in an independent cohort. *The international journal of neuropsychopharmacology* 17, 2069-2074.
- Quelch, D.R., Katsouri, L., Nutt, D.J., Parker, C.A., Tyacke, R.J., 2014. Imaging endogenous opioid peptide release with [¹¹C]carfentanil and [³H]diprenorphine: influence of agonist-induced internalization. *Journal of cerebral blood flow and metabolism : official journal of the International Society of Cerebral Blood Flow and Metabolism* 34, 1604-1612.
- Ray, R., Ruparel, K., Newberg, A., Wileyto, E.P., Loughhead, J.W., Divgi, C., Blendy, J.A., Logan, J., Zubieta, J.K., Lerman, C., 2011. Human Mu Opioid Receptor (OPRM1 A118G) polymorphism is associated with brain mu-opioid receptor binding potential in smokers. *Proceedings of the National Academy of Sciences of the United States of America* 108, 9268-9273.

- Resendez, S.L., Dome, M., Gormley, G., Franco, D., Nevarez, N., Hamid, A.A., Aragona, B.J., 2013. mu-Opioid receptors within subregions of the striatum mediate pair bond formation through parallel yet distinct reward mechanisms. *The Journal of neuroscience : the official journal of the Society for Neuroscience* 33, 9140-9149.
- Schotte, A., Janssen, P.F., Gommeren, W., Luyten, W.H., Van Gompel, P., Lesage, A.S., De Loore, K., Leysen, J.E., 1996. Risperidone compared with new and reference antipsychotic drugs: in vitro and in vivo receptor binding. *Psychopharmacology* 124, 57-73.
- Scott, D.J., Domino, E.F., Heitzeg, M.M., Koeppe, R.A., Ni, L., Guthrie, S., Zubieta, J.K., 2007. Smoking modulation of mu-opioid and dopamine D2 receptor-mediated neurotransmission in humans. *Neuropsychopharmacology : official publication of the American College of Neuropsychopharmacology* 32, 450-457.
- Shapiro, D.A., Renock, S., Arrington, E., Chiodo, L.A., Liu, L.X., Sibley, D.R., Roth, B.L., Mailman, R., 2003. Aripiprazole, a novel atypical antipsychotic drug with a unique and robust pharmacology. *Neuropsychopharmacology : official publication of the American College of Neuropsychopharmacology* 28, 1400-1411.
- Tandon, R., DeQuardo, J.R., Taylor, S.F., McGrath, M., Jibson, M., Eiser, A., Goldman, M., 2000. Phasic and enduring negative symptoms in schizophrenia: biological markers and relationship to outcome. *Schizophrenia research* 45, 191-201.
- Tandon, R., Ribeiro, S.C., DeQuardo, J.R., Goldman, R.S., Goodson, J., Greden, J.F., 1993. Covariance of positive and negative symptoms during neuroleptic treatment in schizophrenia: a replication. *Biological psychiatry* 34, 495-497.
- Tuominen, L., Nummenmaa, L., Keltikangas-Jarvinen, L., Raitakari, O., Hietala, J., 2014. Mapping neurotransmitter networks with PET: an example on serotonin and opioid systems. *Human brain mapping* 35, 1875-1884.
- Veening, J.G., Barendregt, H.P., 2015. The effects of beta-endorphin: state change modification. *Fluids and barriers of the CNS* 12, 3.
- Veening, J.G., Gerrits, P.O., Barendregt, H.P., 2012. Volume transmission of beta-endorphin via the cerebrospinal fluid; a review. *Fluids and barriers of the CNS* 9, 16.
- Volk, D.W., Radchenkova, P.V., Walker, E.M., Sengupta, E.J., Lewis, D.A., 2012. Cortical opioid markers in schizophrenia and across postnatal development. *Cerebral cortex (New York, N.Y. : 1991)* 22, 1215-1223.
- Ward, H.G., Nicklous, D.M., Aloyo, V.J., Simansky, K.J., 2006. Mu-opioid receptor cellular function in the nucleus accumbens is essential for hedonically driven eating. *The European journal of neuroscience* 23, 1605-1613.
- Weerts, E.M., Kim, Y.K., Wand, G.S., Dannals, R.F., Lee, J.S., Frost, J.J., McCaul, M.E., 2008. Differences in delta- and mu-opioid receptor blockade measured by positron emission tomography in naltrexone-treated recently abstinent alcohol-dependent subjects. *Neuropsychopharmacology : official publication of the American College of Neuropsychopharmacology* 33, 653-665.
- Woods, S.W., 2003. Chlorpromazine equivalent doses for the newer atypical antipsychotics. *The Journal of clinical psychiatry* 64, 663-667.

REVIEWERS' COMMENTS:

Reviewer #1 (Remarks to the Author):

The authors have addressed the reviewers comments extensively. One point that was not addressed appropriately was that there is very substantial evidence that alterations in dopamine neurotransmission affect opioid neurotransmission. While the reviewer agrees that antipsychotics do not have significant direct effects on the opioid system, there is a very substantial literature (e.g., S. R. George, M. Kertesz, *Peptides* 8, 487 (1987); E. M. Unterwald, J. M. Rubinfeld, M. J. Kreek, *Neuroreport* 5, 1613 (1994); J. F. Chen, V. J. Aloyo, B. Weiss, *Neuroscience* 54, 669 (1993); H. Steiner, C. R. Gerfen, *Exp. Brain. Res.* 123, 60 (1998) showing that alterations in dopamine neurotransmission affect opioid systems. Hence the potential effect of antipsychotics was potentially under-appreciated.

Reviewer #2 (Remarks to the Author):

The authors did an excellent job in addressing this reviewers' comments. I am particularly happy that the analysis of dorsal and ventral striatum is now included and that the patient characteristics (PANSS scores, smoking status) are now clear.

Please modify the manuscript to better acknowledge the concern raised by Referee #1. Your analysis showing no correlation of antipsychotic dose with MOR availability is welcome but you should acknowledge that the potential for a confound exists as Referee #1 argues.

>> We thank reviewer for raising this issue. We have now amended the manuscript to include this in the discussion:

“Moreover, it is important to note that non-human primate studies show that neither haloperidol nor olanzapine leads to appreciable alterations in the MOR availability, indicating that antipsychotic treatment does not significantly alter MOR1. Thus, it is unlikely that antipsychotic treatment is a significant confounder. Nevertheless, several preclinical studies have shown that alteration in dopaminergic activity can affect the opioidergic neurotransmission. 2-5 Future studies in drug naïve subjects are needed to address this confounding effect. “